# Investigating Language Models for Supporting Complex Group Decisions

## Abstract

Reaching consensus is a central challenge in group decision making as agreement needs to be balanced with diversity of perspectives. Recent AI advances have opened new possibilities for synthesizing complex information and facilitating consensus. We study group decision processes by modeling the complexity of the decision surface, defined by a set of decision problems, each with multiple options. Each solution yields a gain for every participant, and the objective of deliberation is to ensure fairness by equalizing participants' profits. We explore multiple settings: whether gains are private, arbitrary numbers, or ordered sequences; whether the exact gain for each option is public; and whether group communication is expressed in natural language or numerically. Group coordination is facilitated by an AI agent powered by a large language model (LLM). We find that reasoning LLM models perform better than non-reasoning models and that a constraint solver (CPLEX) or a reinforcement learning agent (MCTS) improves the quality of the decision. The performance of reasoning models carries over when the participants rank order their preferences instead of assigning numeric scores. Numeric feedback leads to higher quality solutions than verbal feedback and is also better than when participants state their preference between two decisions. Our findings suggest that while LLMs show promise in facilitating consensus, there remains significant room for improvement in their ability to fully capture and reason over group consensus involving numerical outcomes.

## 1 Introduction

Group decision making is prevalent across domains ranging from business and government to healthcare and science. Reaching consensus is a central challenge in this process, as agreement needs to be balanced with diversity of perspectives. Social psychology research highlights mechanisms of conformity and influence, showing how consensus can emerge but also warning against premature uniformity, such as groupthink (Janis, 1972), or process loss (Steiner, 1972). Models of social choice and negotiation have provided formal frameworks for understanding consensus formation under different voting rules and preference aggregation schemes (Arrow, 1963; Fishburn, 1970). Organizational studies also emphasize structured deliberation and communication norms as means to improve collective outcomes.

Recently, computational and AI approaches have opened new possibilities for improving collective decision processes. Advances in multi-agent systems, social choice modeling, and large language models (LLMs) provide tools for coordinating distributed expertise, synthesizing complex information, and facilitating consensus (Rahwan et al., 2019). These developments highlight the importance of group decision making as well as the transformative potential of AI to support such complex processes at scale.

Reaching a consensus through negotiation and compromise that satisfies all parties involved is often a challenge. In situations where decision problems become more complex, it is difficult to identify the optimal solution, necessitating rigorous computation and in-depth discussions. Consequently, numerous researchers have explored the potential of computational methods to assist humans in group decision-making tasks(Sun et al., 2025; Jia et al., 2024).

Our specific decision problem is motivated by three considerations. First, linear (or integer) programming is essential in decision making as it provides a rigorous, quantitative method for choosing

the best possible course of action under constraints. By expressing objectives and encoding real-world limitations as inequalities, many potential solutions can be evaluated to identify the optimal one. This generic characterization is especially useful in complex environments such as supply chains, scheduling, finance, energy systems, and public policy. Second, integer programming is a natural generalization of linear programming that is NP-hard. This means that any algorithm (or human being) can at best produce an approximation. So, even in environments where tools may assist LLMs, approximations to the ideal solution can be compared for how well they satisfy the objective and how efficient they are. Finally, participants in decision making may hold private preferences that are not shared with others, and the only hint that the group may have is their degree of satisfaction in a natural language response. During the interaction process, we expect LLMs to infer participants' underlying preferences and reason and search for a solution that is as fair as possible for all participants. This is reminiscent of the hidden profile problem in psychology where groups systematically fail to share and integrate unique (unshared) information, leading them to make inferior decisions, even though the correct choice would be obvious if all information were pooled (Stasser & Titus, 1985; Lu et al., 2012; Pescetelli et al., 2022).

We study group decision processes considering the complexity of the decision surface. The more rugged and high-dimensional the space becomes, the harder the search for optimality will be. In our setting, the decision surface is defined by a set of decision problems, each of which has a number of options for solution. Each solution accords a gain for a participant in the decision. The goal of deliberations is to ensure that the set of decisions is fair in that the profit for each participant is equal. We consider different settings whether the gains are private, arbitrary, or an ordered sequence, whether the exact gain for a set of choices is public, or whether the group communicates in a natural language or numerically.

The coordination of the group is carried out by an AI agent modeled by an LLM. We consider different kinds of reasoning and non-reasoning LLMs. We also augment the LLM with a constraint solver such as CPLEX and investigate the use of a reinforcement learning agent based on Monte Carlo Tree Searching (MCTS) (Browne et al., 2012). The principal evaluation metric of interest is the quality of the final answer.

Our main findings are as follows.

- Reasoning models perform better than non-reasoning models for complex decisions. The performance of reasoning models carries over when the participants rank order their preferences instead of assigning numeric scores. (Section 5.1)
- MCTS is better than the sole use of LLM for both numeric and verbal feedback. (Section 5.2)
- Numeric feedback is better than verbal feedback and comparative feedback. (Section 5.3 and Section 5.4)
- A constraint solver improves the quality of the decision, especially when the decision surface becomes complex. (Section 5.5)

## 2 RELATED WORK

Group decision making has long been a central topic of interest, with researchers investigating how computational methods can simulate and assist human collective decisions. Naturally, multi-agent systems, as a tool for modeling group behaviors, offer unique advantages for addressing group decision problems. A considerable body of work has focused on designing more accurate agents to better capture collective dynamics (Wanyama & Far, 2007; Yager, 2002b; Xuan et al., 2001). For example, Yager (2002a) modeled group preference functions in a mathematical framework, and Marreiros et al. (2008) introduced ABS4GD, a multi-agent system that simulates group decision processes while incorporating emotional and argumentative aspects. However, due to the limited reasoning capacity of traditional models, applying multi-agent systems to group decision problems remained largely confined to idealized settings and struggled to adapt to real-world complexity. In particular, they faced significant challenges in accurately quantifying individual preferences.

With the advent of LLMs, researchers began probing whether language-based agents can represent people and whether they can improve human deliberation. Generative agents showed that LLM-

driven agents embedded in a sandbox environment produce believable routines and emergent social behavior, suggesting that simulated populations can sustain coherent, multi-day social dynamics (Park et al., 2023). In parallel, economics and computational social science introduced "homo silicus" to study strategic behavior with AI-only agents, including whether populations of LLMs reproduce human-like phenomena such as trust, reciprocity, and coalition formation (Filippas et al., 2024; Huang et al., 2024). Beyond simulation, LLMs have been deployed as process interventions: a "devil's advocate" that injects counterarguments can improve groups' critical engagement with AI suggestions, while mediation prompts can depolarize heated debates by illuminating areas of agreement (Chiang et al., 2024; Govers et al., 2024).

Using LLMs to solve mathematical problems has become a recent research focus. Chain-of-Thought (CoT) markedly improves performance on complex questions; Self-Consistency stabilizes answers by sampling multiple reasoning paths and taking a majority vote (Wei et al., 2022). Current research has shifted to process supervision and verification: 'Let's Verify Step by Step' trains process rewards with stepwise feedback (Lightman et al., 2023); Math-shepherd and Math-Minos demonstrate that process-level scoring and natural-language feedback significantly improve correctness and controllability (Wang et al., 2023; Gao et al., 2024); and domain models such as DeepSeekMath Shao et al. (2024) and Qwen2.5-Math (Yang et al., 2024) leverage large-scale mathematical corpora and self-improvement schemes to push open-source systems toward performance comparable to closed-source models on challenging benchmarks.

Recent studies have explored the use of Monte Carlo Tree Search (MCTS) for solving mixed-integer linear programming (MILP) and other combinatorial optimization problems, demonstrating its effectiveness as a search-based heuristic (Silva et al., 2022; Zhang & Peng, 2024). Beyond classical MIP settings, MCTS has also been enhanced with local optimization and gradient-based refinements to accelerate convergence and improve performance in continuous or partially observable decision spaces (Zhai & Gao, 2022; Lev-Yehudi et al., 2025). More recently, Ren et al. (2025) integrates LLM-generated guidance into the search process, though its focus lies in information retrieval rather than decision optimization. Taken together, these studies highlight the promise of MCTS for structured optimization, but most assume fully specified numerical formulations and lack integration with natural-language constraints or uncertain variables. Our work addresses this gap by combining MCTS with language-based feedback and partially observed parameters, enabling decision-making in settings where problem specifications are noisy, implicit, or communicated in natural language.

## 3 PROBLEM DEFINITION

The specific optimization that a team needs to solve is a mixed-integer programming problem. Each agent $i$ (group participant) evaluates an option $j_k$ for decision $j$ with a gain $g_{ijk}$. Let $V_i$ represent the total value accorded to agent $i$.

**Objective function:**

$$\text{Minimize} \quad (high - low)$$

**Constraints for agent $i$:**

$$g_{ijk} \geq 0 \quad \forall j, k,; \ V_i = \sum_j \sum_k g_{ijk} x_{jk}; \ high \geq V_i; \ low \leq V_i$$

**Constraints for decision $j$:**

$$x_{jk} \geq 0 \quad \forall k; \ \sum_k x_{jk} = 1; \ x_{jk} \in \{0, 1\} \quad \forall k$$

For each participant, we define *high* as the highest value the participant can obtain across all available options, and *low* as the lowest value. We refer to the difference between the value of the options chosen and the best possible value for a participant as the participant's *divergence*. The goal of the optimizer is to minimize the maximum divergence across the participants. Any random choice of options results in a possible value of the maximum divergence, termed the divergence of the solution. We use it to evaluate the quality of a solution. Note that if the binary constraints on the

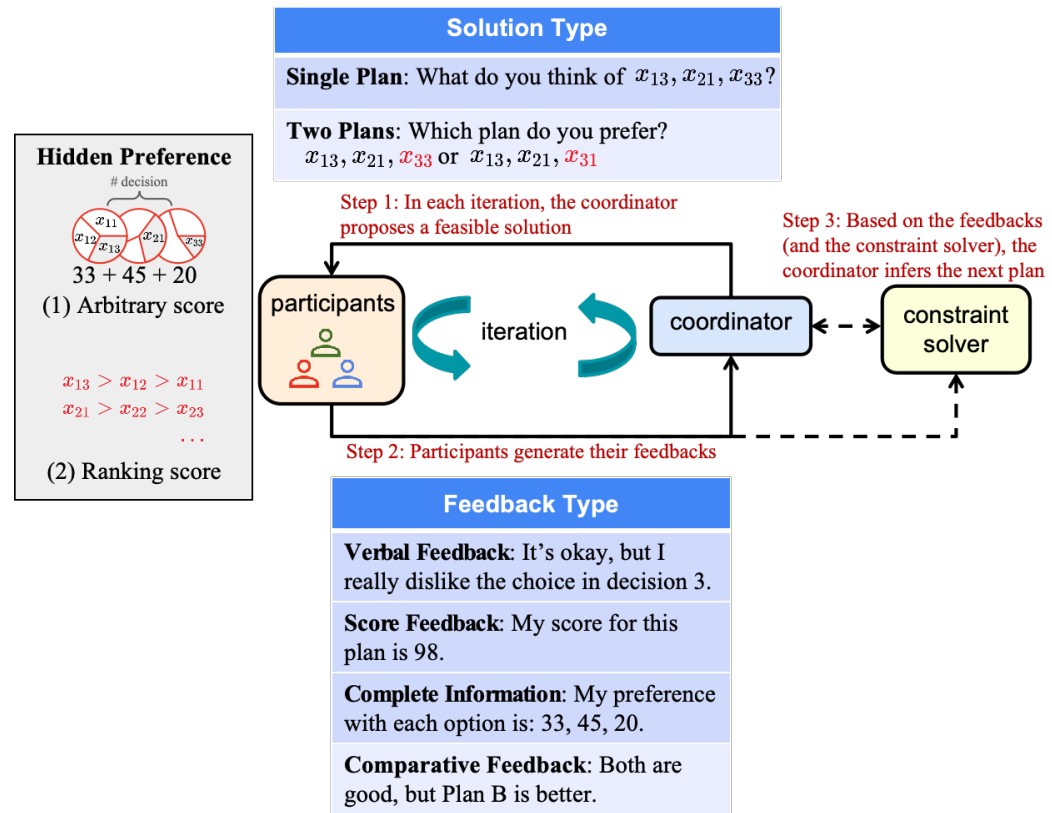

Figure 1: Experimental setup for evaluating different combinations of environment configurations. The varying factors include the type of participants' hidden preferences, the type of feedback provided, and the output format of the coordinator. When the hidden preference type is arbitrary score, we compare verbal feedback, score feedback, and complete information. When the hidden preference type is ranking-based score, we evaluate verbal feedback. We also test the case where the coordinator produces two plans, in which participants provide comparative feedback.

options for a decision are weakened to continuous values (which still sum to one), then we obtain an LP formulation.

# 4 EXPERIMENTAL SETTING

We design a series of experimental settings to systematically evaluate the capability of LLMs as decision makers in multi-agent group decision-making tasks.

## 4.1 GENERAL FRAMEWORK

Each problem consists of $n$ decisions, where each decision is associated with $m$ options. For every decision/option configuration, we generate 30 different problem instances. In each problem, three participants are assigned individual preferences over the options. An LLM-based planner then iteratively interacts with the participants for up to five rounds in order to produce a final solution. Both the planner and the participants are instantiated as LLMs: the planner varies across models depending on the experiment, while the participants generating verbal feedback are uniformly instantiated with GPT-4o-mini, given its relatively simple and consistent role. For the algorithmic details, we provide a comprehensive description in the appendix C

## 4.2 EVALUATION METRIC

To measure performance, we use the average *divergence* across problems. For a given plan, divergence is defined as the maximum difference among the three participants' total preference scores over the selected options. A smaller divergence indicates that the participants are more aligned in their satisfaction with the plan, reflecting a fairer decision.

## 4.3 EXPERIMENTAL VARIANTS

We consider three major types of experimental variants, as illustrated in Figure 1.

**Arbitrary Scores.** Preferences are represented as arbitrary scores, with each decision's options sharing 100 points.

- **Verbal Feedback:** Each participant verbalizes their preference in natural language and conveys this to the planner. This represents the highest difficulty setting.

- **Score Feedback:** Participants only report the overall score of the current plan without revealing detailed scores. This corresponds to the standard mixed-integer programming formulation.

- **Complete Information:** The planner is given exact preferences of each participant directly, without any need for additional communication.

**Ranking-based Scores.** Each participant only maintains a relative ranking of options within a decision. Participants then produce verbal feedback reflecting this ranking. Divergence is computed based on the induced ordering.

**Plan Comparison.** Instead of proposing a single candidate plan, the planner outputs two plans at each iteration. Participants compare the two and provide feedback, allowing for more intuitive and direct signals.

## 5 FINDINGS

### 5.1 REASONING IS CRUCIAL IN COMPLEX DECISION MAKING

To assess how different models understand mixed-integer programming, we conducted experiments on several representative systems. We varied the number of decisions to examine its effect on performance. For each configuration, we randomly generated 30 distinct problem instances and compared the results for 3-30 and 35 decisions. Our primary objective was to contrast reasoning-enabled models with non-reasoning models. For the former, we used GPT-o3-mini, Gemini-2.5-Flash, and DeepSeek-R1; for the latter, we used DeepSeek-V3 and a variant of Gemini 2.5 Flash with the thinking step disabled. The results are shown in Figure 2. Each marker denotes the mean over 30 trials. Overall, the reasoning-enabled models——particularly o3-mini and Gemini-2.5-Flash—— substantially outperform the rest. By contrast, all non-reasoning variants perform close to tran-dom, suggesting that without an explicit reasoning procedure the models tend to return near-random solutions and fail to reliably capture the task context. Furthermore, comparing Gemini-2.5-Flash with and without the thinking mode indicates that explicit reasoning is crucial for understanding mixed-integer programming: without thinking enabled, Gemini-2.5-Flash behaves almost randomly, whereas enabling it yields performance on par with o3-mini. Since CPLEX requires access to complete information, we treat its result as an upper bound. In the divergence metric, values closer to 0 indicate a more equitable solution. As shown, CPLEX is able to find the optimal solution in the vast majority of cases.

We also investigate the LLMs' ability to interpret rankings instead of mapping text into distributional values. In this experimental setup, each option is assigned a specific preference rank rather than a numerical score. That is, given n options, each is assigned a ranking from 1 to n, with larger values indicating stronger preference. We evaluated five models with option numbers ranging from 4 to

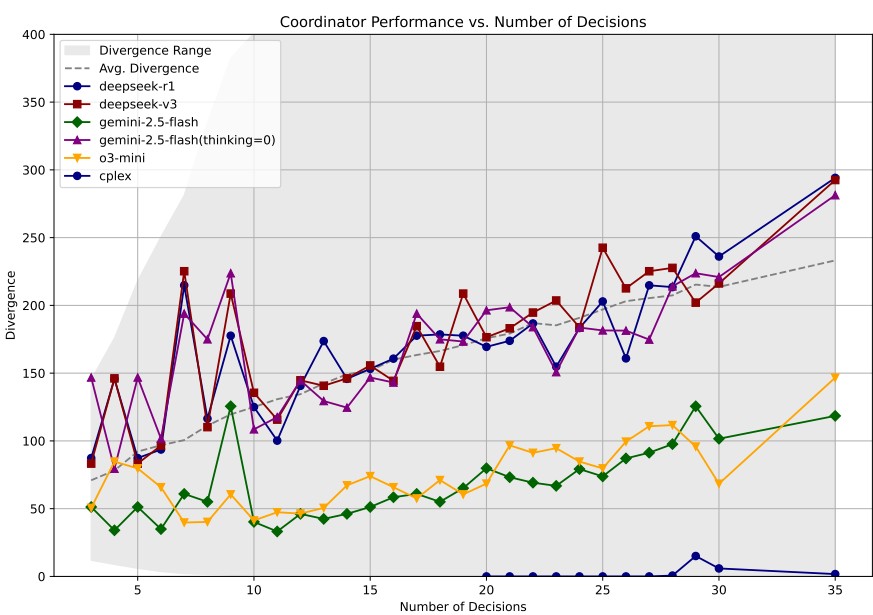

Figure 2: Comparison of models with complete information (3—30 and 35 decisions, 3 options). Most reasoning models perform significantly better than random, while all non-reasoning models behave close to random.

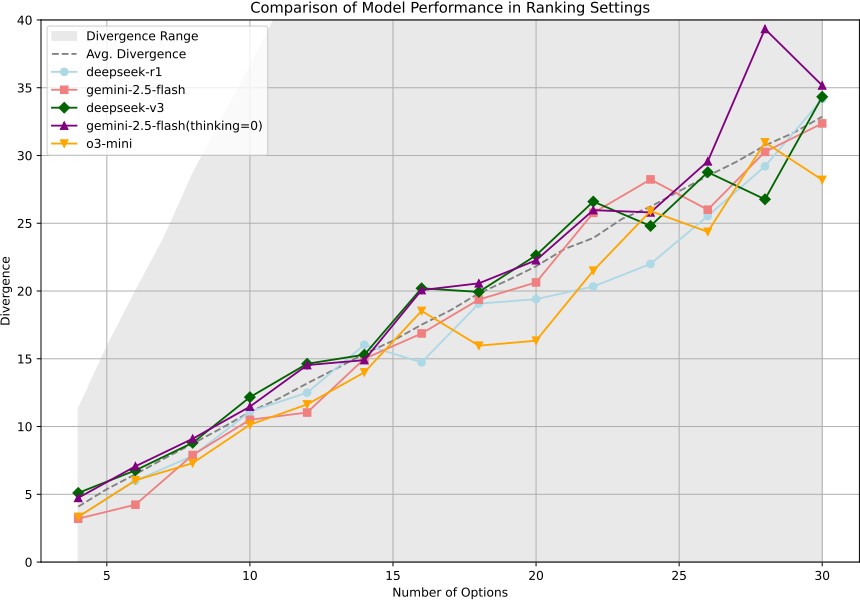

Figure 3: Results of ranking-based feedback with five models (5 decisions, 4–30 options in steps of 2). Reasoning models (GPT-o3-mini, DeepSeek-R1, Gemini-2.5-Flash) consistently outperform random, whereas non-reasoning models remain less stable.

30 in steps of 2, with a fixed number of 5 decisions. For reasoning models, we tested DeepSeek-R1, Gemini-2.5-Flash, and GPT-o3-mini; for non-reasoning models, we tested DeepSeek-V3 and Gemini-2.5-Flash without the reasoning step.

As shown in Figure 3, GPT-o3-mini and DeepSeek-R1 achieved the best overall performance. By contrast, non-reasoning models generally hovered around random, with DeepSeek-V3 performing slightly better than Gemini-2.5-Flash without reasoning in certain cases. Under the ranking-based setting, reasoning models consistently outperformed the average even in high-dimensional spaces, while non-reasoning models exhibited less stable behavior.

## 5.2 MCTS Performs Better than Vanilla LLM

LLMs primarily rely on heuristic search to identify promising solutions. However, this approach is susceptible to becoming trapped in local optima, failing to discover the global optimum. To address this limitation, we integrate MCTS as a coordinator agent and evaluate whether it can outperform the vanilla LLM approach.

To apply MCTS, we model the decision process as a search tree in which each node corresponds to an option and each level represents a decision step. A path from the root to a leaf thus encodes a complete plan. MCTS iteratively samples such paths, evaluates their rewards, and updates its statistics to guide subsequent exploration. The algorithmic details are provided in the Appendix B. We repeat this process for 5 iterations and compare the resulting plans against those generated by using only the LLM (Gemini-2.5-Flash) as the coordinator. This comparison allows us to isolate the contribution of the tree-search mechanism to the overall decision quality. As shown in Figure 4, MCTS outperforms the LLM in most cases, both in the score-feedback and verbal-feedback settings.

We observe that in the scenario with three decisions and three options, the performance of MCTS is worse than the average. This degradation can be attributed to the LLM's limited accuracy in estimating numeric values from verbal feedback, which propagates errors into the MCTS search process and ultimately biases it toward unfair solutions.

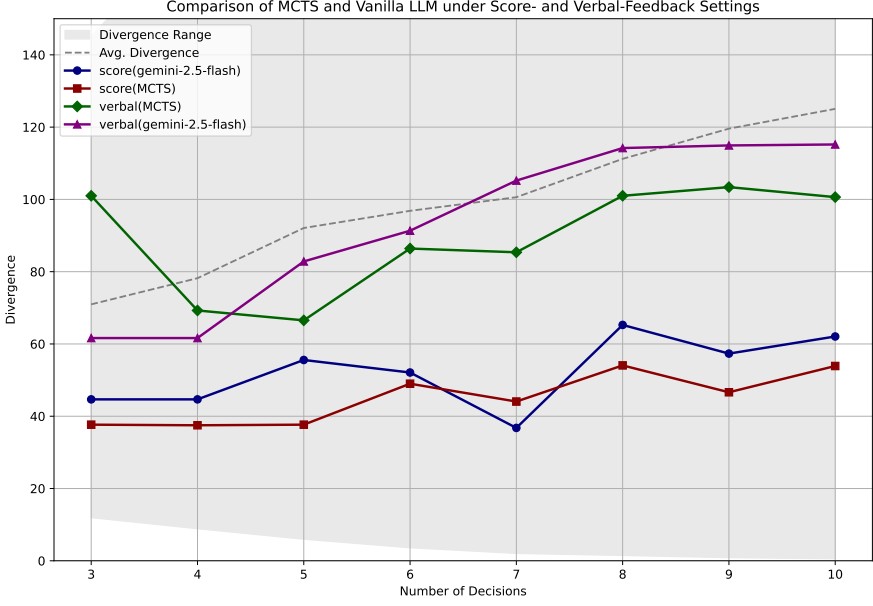

Figure 4: Comparison of MCTS and Gemini-2.5-Flash under score feedback and verbal feedback settings (3–10 decisions, 3 options). Using MCTS as the coordinator consistently yields better outcomes than using the LLM alone, regardless of feedback type, given the same number of iterations.

### 5.3 SCORE FEEDBACK OUTPERFORMS VERBAL FEEDBACK

Translating arbitrary numbers into text and then back to numbers for optimization is challenging. We conducted comparative experiments on Gemini-2.5-Flash, testing decision numbers ranging from 3 to 10 with 3 options per decision. In the score feedback setting, the LLM was given the sum of the preference scores for all options within a plan. In the verbal feedback setting, we simulated three distinct participants, each aware of their own preference score, who generated short verbal feedback based on these scores. The LLM could only infer participant satisfaction from these verbal descriptions.

As shown in Figure 4, the verbal feedback group generally performed closer to random, and in cases with 7 or 8 decisions, even worse than the baseline. By contrast, when the LLM was given explicit preference scores for each participant, it was able to maintain a better divergence. These findings highlight that while LLMs can reason in the context of mixed-integer programming, grounding natural language feedback into precise numerical scores remains a significant challenge.

### 5.4 SCORE FEEDBACK OUTPERFORMS COMPARATIVE FEEDBACK

To examine whether alternative feedback mechanisms can enhance an LLM's ability to capture participants' implicit preferences, we compared score feedback with a pairwise comparative feedback setting. In the latter, the LLM was asked to propose two candidate plans simultaneously, and participants expressed their preferences by comparing the two, rather than evaluating a single plan in isolation. Compared with the earlier verbal feedback setting, this form of feedback is more direct, since it conveys judgments over two options at once.

We evaluated Gemini-2.5-Flash with decision numbers ranging from 3 to 10 and 3 options per decision. As shown in Figure 5, while comparative feedback yielded better results than single-plan verbal feedback in higher-dimensional settings, score feedback still demonstrated a clear and consistent advantage. This indicates that for LLMs, the ability to extract precise numerical scores from textual feedback remains a key bottleneck.

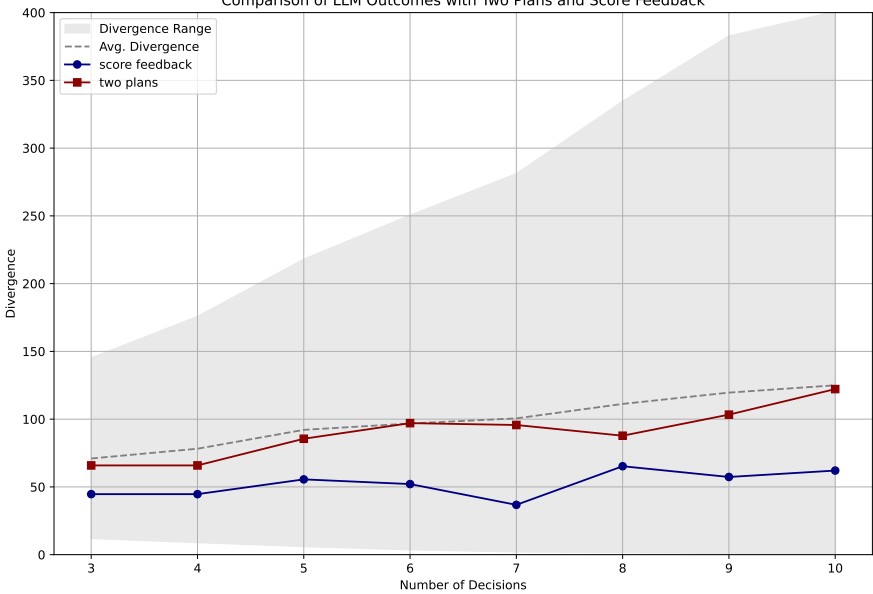

Figure 5: While comparative feedback offers some improvement, score feedback is consistently more accurate than verbal feedback.

## 5.5 CONSTRAINT SOLVER IMPROVES PERFORMANCE

In high-dimensional spaces, LLMs struggle to maintain consistent performance. Incorporating external knowledge can help guide their decisions. We employed CPLEX, a classical optimization solver, as an auxiliary tool and evaluated its effect on GPT-o3-mini with 20–30 and 35 decisions. This experiment was done for the complete information setting. As shown in Figure 6, without CPLEX, the LLM's divergence increased steadily with the number of decisions. By contrast, when guided by CPLEX, the coordinator can directly identify the global optimum in most cases. However, this approach requires CPLEX to have access to every participant's score for every option, which is rarely feasible in real-world scenarios.

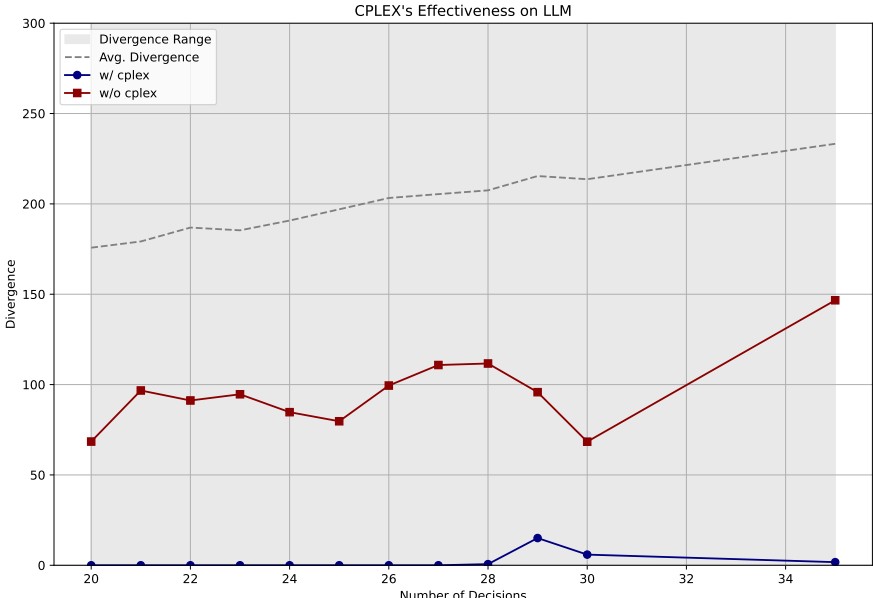

Figure 6: Performance of GPT-o3-mini with and without CPLEX assistance (20–30 and 35 decisions, 3 options). CPLEX consistently reduces divergence and improves performance.

## 5.6 METRIC-SPECIFIC PREFERENCES IN LLM SEARCH PROCESSES

In the previous section, we focused on divergence, which serves as a measure of fairness. This metric is analogous to envy-fairness, aiming to identify plans under which all participants achieve comparable levels of satisfaction. Various other metrics can also be used to evaluate the quality of a proposed solution.

We implemented another objective function, group welfare, motivated by Harsanyi (1955). In our setting, welfare is defined as the sum of all participants' individual gains under a given plan. Accordingly, we updated the LLM coordinator's prompt to explicitly instruct the planner to search for solutions that maximize the total group benefit. Under this setup, we compared the performance of score feedback and verbal feedback strategies. The results show that when the decision dimensionality is low, both strategies perform similarly. However, as the number of decisions increases and the search space expands, verbal feedback consistently outperforms score feedback, achieving substantially higher welfare.

Through detailed case analysis, we find that this performance gap stems from the fundamentally different search behaviors induced by the two feedback modalities. When receiving score feedback, the LLM coordinator tends to adopt a greedy and conservative local search strategy, typically modifying only one or two decisions at a time and reverting changes that lower the welfare. This risk-averse

behavior makes it more prone to getting trapped in local optima in high-dimensional search spaces. In contrast, verbal feedback provides richer contextual information (e.g., which options are "positively received" or "consistently negative"), enabling the coordinator to perform more assertive and globally informed exploration by simultaneously altering multiple decisions or introducing previously untested options. As a result, verbal feedback is more capable of escaping local optima and identifying higher-welfare solutions when the decision space grows.

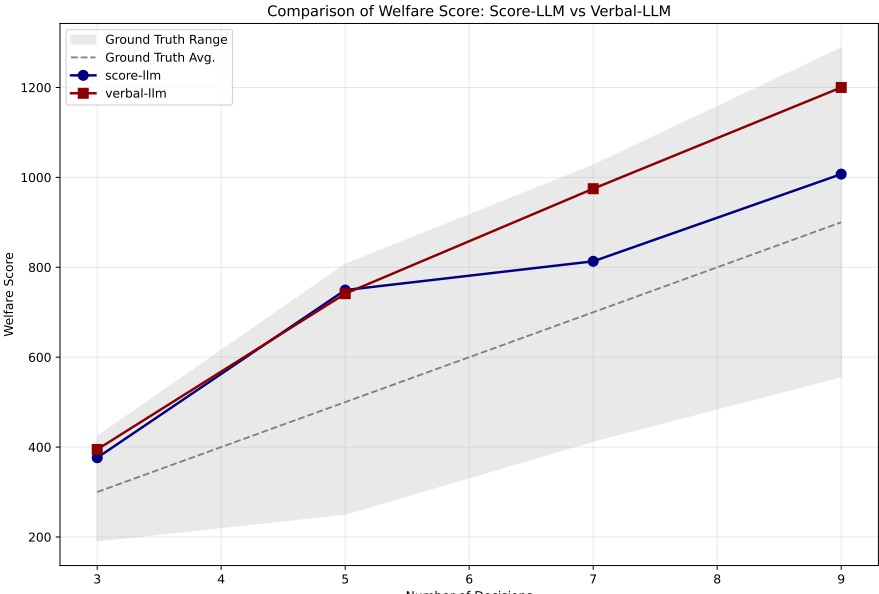

Figure 7: Performance of Gemini-2.5-Flash under score feedback and verbal feedback across settings with 3, 5, 7, and 9 decisions and 3 options. Verbal feedback maintains strong performance even as the number of decisions increases, whereas score feedback degrades substantially in higher-dimensional settings.

## 6 CONCLUSION AND FUTURE WORK

We defined several group decision-making settings and examined how language models can assist in identifying fair decisions both when individual preferences are hidden and under an idealized full-information regime. Across these settings, we systematically compared reasoning vs. non-reasoning models, search-based coordination via MCTS, feedback modalities (numeric, verbal, and comparative), and the integration of a constraint solver. Our study yields four consistent findings: (i) reasoning models outperform non-reasoning models for complex decisions, and this advantage persists when preferences are expressed as rankings rather than numeric scores; (ii) MCTS-based coordination improves outcomes over using an LLM alone under both numeric and verbal feedback; (iii) numeric feedback leads to higher-quality solutions than verbal or comparative feedback; and (iv) a constraint solver further boosts decision quality, especially as the decision surface becomes more combinatorial.

Several important directions remain open. First, our current setting assumes that all participants behave honestly and cooperatively. Second, our evaluation relies solely on simulated LLM participants. Human-subject studies are needed to understand how these methods perform in real-world group collaboration and to assess their effect on human decision-making dynamics. Future work could explore model distillation, fine-tuning, or lightweight architectures that enable faster inference and real-time coordination without sacrificing decision quality.

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

## A  PROMPTS AND EXAMPLES

The following is the prompt used by the agent to generate verbal feedback.

```
You are {player_name}, a player with private preference scores over
    options for each meal.
For each meal's options, you have distributed a total of 100 points,
    where more points mean stronger preference.
Your task is to provide a short, natural-language feedback about how
    satisfied you are with each option in the proposed plan, based on
    your scores.
Do not reveal your numeric scores.
Keep it concise (<=40 words) and reference each meal briefly.

Your private scores for all options are: {all_scores_text}
Proposed plan is: {plan_text}
Your latent numeric satisfaction for the selected options (from your
    private scores, where higher is better, the highest score is 100): {
    score_text}
Write your feedback now.
```

Table 1 presents an example with three decisions and three options per decision. It reports a single participant's preference scores, the plan produced by the coordinator in one round, and the corresponding feedback.

Table 1: Example of a single player's preference scores, coordinator plan, and corresponding feedback for a scenario with three decisions and three options per decision.

| Decision | Meal 1 | Meal 2 | Meal 3 |
|---|---|---|---|
| **Options** | Option 1 / 2 / 3 | Option 1 / 2 / 3 | Option 1 / 2 / 3 |
| **Preference Score** | 80 / 10 / 10 | 50 / 30 / 20 | 25 / 30 / 45 |
| **Plan** | Option 1 | Option 2 | Option 2 |
| **Verbal Feedback** | *I really enjoy Meal 1, but Meal 2 could be improved. Meal 3 is acceptable.* | | |

The following is the prompt used by the LLM coordinator with complete information setting.

```
You are a team coordinator assisting in a team decision-making task.
The team needs to select a menu for n meals, with m options available for
    each meal.
Your goal is to select a plan based on the total rating scores from each
    player, choosing the plan where the score differences among the p
    players are minimized. In other words, you need to find the fairest
    plan. You can only search up to r times, so you should aim to find
    the minimal solution when the number of KNOWN PLANS is less than r.

Rules:
1. Provide a plan. Output in JSON format: {{"plan": [option...]}}. Do not
    output anything else, and do not use Markdown format.
2. For each known plan, calculate the total score for each player. The
    maximum difference among the three players' scores is defined as the
    fairness score of that plan. When the number of known plans is {round
    -1}, select the one with the smallest fairness score among the known
    plans as the final answer. Output in JSON format: {{"plan": [meal
    options with the smallest score difference], "reason": "the reason
    why it is the optimal solution", "status": "FINISH"}}. Do not output
    anything else, and do not use Markdown format.
3. If the score difference is not small enough and further search is
    needed, then based on all known information, propose a new plan that
    minimizes the maximum score difference among the players. Output in
    JSON format: {{"plan": [option which is not in the KNOWN PLANS], "
    reason": "What options did you change to make the entire plan fairer,
```

```
    and why did you choose to change that part?", "status": "RUNNING"
    }}. Do not output anything else, and do not use Markdown format.

Example (3 players):
plan1: apple, milk, bread
scores:
- player1: 3, 30, 50
- player2: 80, 70, 60
- player3: 35, 1, 45

plan2: banana, milk, bread
scores:
- player1: 70, 30, 50
- player2: 5, 70, 60
- player3: 25, 1, 45

Analysis: In Plan 1, the total scores for different players are
    3+30+50=83, 80+70+60=210, and 31+1+45=77. Although Player 2 is
    relatively satisfied, Player 1 and Player 3 are not, resulting in a
    fairness divergence calculated as the score of the most satisfied
    player minus that of the least satisfied one: 210 - 77 = 133. In Plan
     2, "apple" is replaced with "banana", and the total scores for the
    three players change to 150, 135, and 71, respectively. The fairness
    divergence is now 150 - 71 = 79, making Plan 2 better than Plan 1.
    However, since the divergence is still 79, the plan needs further
    improvement. As Player 3 is not very satisfied with either "banana"
    or "milk" in Plan 2, the next step is to try replacing "milk" with
    another option to see if the difference can be reduced.

Options: {all options}

Known plans: {known plans}

Current plan is {...}. What's the next fairer plan? Output in JSON format
    : {{'plan': [{meals} option which is not in the KNOWN PLANS], 'reason
    ': 'What options did you change to make the entire plan fairer, and
    why did you choose to change that part?', 'status': 'RUNNING'}}. DO
    NOT use markdown format. DO NOT output any other explanation or
    analysis.
```

# B    ALGORITHMIC DETAILS OF MCTS

## B.1    MOTIVATION

In our setting, the reward (negative divergence) of a plan is only revealed after a complete plan is proposed. To efficiently explore the combinatorial search space, we adopt the PUCT tree policy introduced in AlphaGo (Silver et al., 2016) and refined in AlphaZero (Silver et al., 2017). PUCT combines exploitation ($Q$-values) with an exploration term weighted by a prior $P(a \mid s)$, biasing the search toward promising actions while still ensuring exploration:

$$a = \arg \max_{b \in \mathcal{A}(s)} \Big[ Q(s,b) + c_{\mathrm{puct}} P(b \mid s) \frac{\sqrt{\sum N(s,b)}}{1 + N(s,b)} \Big],$$

where $\mathcal{A}(s)$ is the set of legal actions available at state $s$, $N(s,a)$ are visit counts, and $P(b \mid s)$ are priors (initialized uniform). At the root node, we additionally mix Dirichlet noise into the prior (Silver et al., 2017):

$$P'(b \mid s_0) = (1 - \varepsilon) P(b \mid s_0) + \varepsilon \eta_b, \qquad \eta \sim \mathrm{Dir}(\alpha),$$

which encourages early exploration of multiple root actions and prevents search collapse.

## B.2    ALGORITHM

Our MCTS variant does not use rollouts or heuristic value estimates. Each search round proceeds as follows:

1. **Selection:** Start from the root $s_0$ and select actions according to the PUCT rule until reaching a leaf or an unexpanded state.

2. **Expansion:** Expand the new state by enumerating legal actions and initializing uniform priors (mixing Dirichlet noise at the root).

3. **Evaluation:** When a terminal state $\pi$ is reached, query the environment oracle for its divergence $\mathrm{div}(\pi)$ and set reward $r = -\mathrm{div}(\pi)$.

4. **Backpropagation:** Update $N(s,a)$, $W(s,a)$, and $Q(s,a)$ along the traversed path and record $\pi^\star$ if it achieves a new minimum divergence.

The algorithm maintains $(\pi^\star, \mathrm{UB}^\star)$, the best plan and divergence observed so far, which are updated after each round.

---

**Algorithm 1** Black-box MCTS

---

**Input:** Environment $\mathcal{E}$, exploration constant $c_{\mathrm{puct}}$, root noise $(\varepsilon, \alpha)$
**Output:** Best plan $\pi^\star$ and divergence $\mathrm{UB}^\star$

---

$s \leftarrow \mathsf{initial}()$; expand if needed and set $P(\cdot \mid s)$ uniform.
Mix root noise $P'(\cdot \mid s_0) = (1 - \varepsilon)P + \varepsilon \mathrm{Dir}(\alpha)$.
**while** $s$ not terminal **do**
  select $a$ by PUCT: $a = \arg \max_{b \in \mathcal{A}(s)} \big[ Q(s,b) + c_{\mathrm{puct}} P(b|s) \frac{\sqrt{\sum N(s,b)}}{1 + N(s,b)} \big]$.
  $s \leftarrow \mathsf{step}(s,a)$.
**end while**
Query oracle for divergence $\mathrm{div}(\pi)$, set $r = -\mathrm{div}(\pi)$.
Backpropagate $r$ along path, update $N, W, Q$ and best $(\pi^\star, \mathrm{UB}^\star)$.

---

## B.3    HYPERPARAMETERS

We set $c_{\mathrm{puct}} = 1.5$ to balance exploration and exploitation. Root noise uses $\varepsilon = 0.25$ and $\alpha = 0.3$, following Silver et al. (2017). At the root, we apply temperature sampling with $\tau = 1.0$ and restrict to the top-$K = 2$ actions before sampling to encourage diversity in early rounds. These hyperparameters were chosen empirically to achieve fast convergence without sacrificing coverage of alternative plans.

## C  Overall Algorithmic Framework

This appendix outlines the overall algorithmic framework used across all experiments. Although specific settings (e.g., feedback type or constraint solver usage) may slightly alter the feedback signal $f$ or the gain estimation procedure, the process consistently follows the same iterative structure. The purpose of this framework is to assess the LLM coordinator's ability to understand participants' preferences and reason over the group decision problem in a multi-agent setting. Detailed procedural steps are provided below.

---

**Algorithm 2** Coordinator Searching Process

---

1: Initialize initial solution $s_1$ from LLM coordinator
2: Initialize global gain history $\mathcal{G} \leftarrow \emptyset$
3: **for** $i = 1$ to $T$ **do**
4:     **for** each participant $p \in \{1, 2, 3\}$ **do**
5:         $f_i^{(p)} \leftarrow \text{Eval}_p(s_i)$
6:     **end for**
7:     $g_i \leftarrow \text{EstimateGain}(s_i, f_i^{(1)}, f_i^{(2)}, f_i^{(3)})$
8:     $\mathcal{G} \leftarrow \mathcal{G} \cup \{g_i\}$
9:     **if** constraint solver is needed **then**
10:         $s_{i+1} \leftarrow \text{ConstraintSolver}(s_i, g_i, \mathcal{G})$
11:     **end if**
12:     $s_{i+1} \leftarrow \text{Search}(s_{i+1})$
13:     **if** coordinator decides to stop **then**
14:         **break**
15:     **end if**
16: **end for**
17: **return** $s_i$

---

**Description.** At iteration $i$, the LLM coordinator maintains a candidate group solution $s_i = \{o_i^{(1)}, o_i^{(2)}, \ldots, o_i^{(n)}\}$, where $o_i^{(d)}$ denotes the option chosen for decision $d$. Each participant $p \in \{1, 2, 3\}$ evaluates the current solution according to its (implicit) preference function $\text{Eval}_p(\cdot)$ and produces a feedback signal $f_i^{(p)}$ (either numeric scores or verbal feedback). These feedback signals are transformed into an estimated gain vector $g_i = \text{EstimateGain}(s_i, f_i^{(1)}, f_i^{(2)}, f_i^{(3)})$, which serves as the quantitative input to the constraint solver.

The constraint solver can be instantiated as either MCTS or CPLEX. For MCTS, the suggestion $s_{i+1}$ is obtained by exploring local modifications around $s_i$ guided by the estimated gain $g_i$. For CPLEX, which requires a complete gain matrix, the algorithm maintains a global history $\mathcal{G}$ of estimated gains across iterations and constructs an aggregated optimization input from $\mathcal{G}$. Under the assumption that the accumulated gains approximate the true preferences, the solution returned by CPLEX can be interpreted as an upper-bound estimate of the achievable envy-free solution. In both cases, the LLM coordinator refines the solver suggestion $\text{Search}(\cdot)$ into the next solution $s_{i+1}$ and decides whether to continue or terminate the search. The procedure stops either when the coordinator explicitly chooses to terminate or when the maximum number of iterations $T$ is reached.

## D  Failure Case Analysis

To better understand the limitations of different model types in the group decision-making task, we analyze representative failure cases from both non-reasoning and reasoning models. We focus on how the search trajectory, intermediate justifications, and divergence patterns contribute to suboptimal outcomes.

### D.1  Non-Reasoning Models

Overall, the non-reasoning model fails because it can recognize the need to search over alternative plans, but it does not perform targeted search with respect to the divergence metric. The model

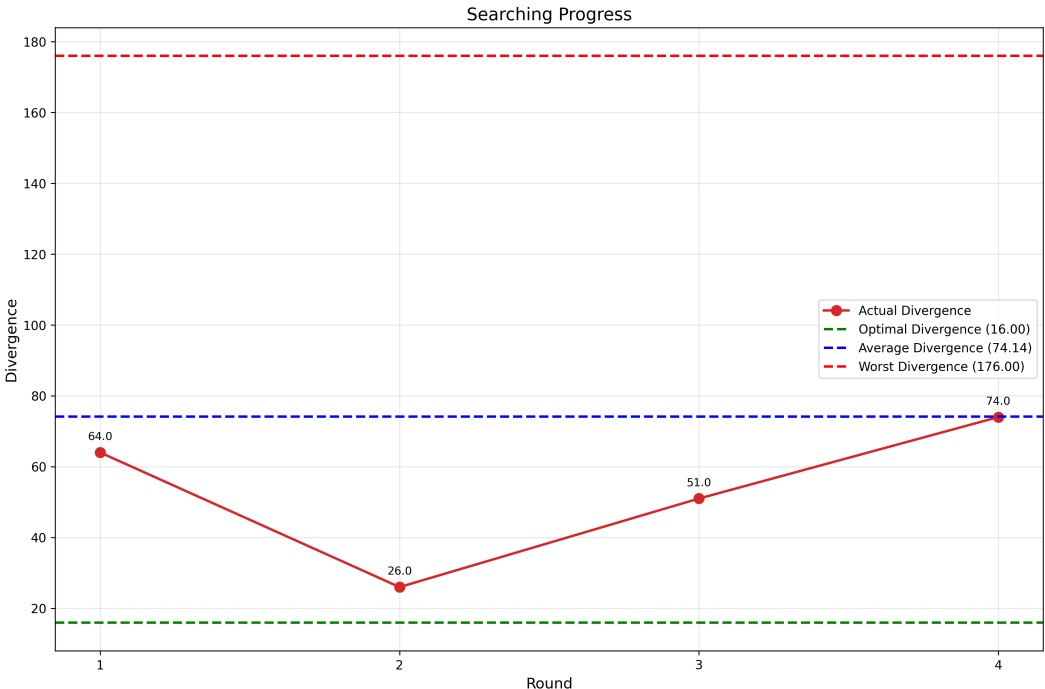

Figure 8: Search trajectory of the non-reasoning model in the 4-decision, 3-option setting.

focuses almost exclusively on locally increasing the weakest participant's score (participant3) by modifying one meal at a time, without consistently monitoring how these changes affect the global divergence. As a result, it unintentionally moves away from the best plan: after reaching a low-divergence solution in Round 2, subsequent updates in Rounds 3–5 repeatedly worsen fairness (higher divergence), even though the textual justifications claim to "reduce the difference." This behavior shows that, without explicit reasoning or metric-aware guidance, the model can search, but it cannot reliably search for lower divergence.

| | Plan | Scores (p1, p2, p3) | Reasoning Summary |
|---|---|---|---|
| 1 | [option1, option1, option1, option1] | (169, 157, 105) | Uses all option1s as a baseline plan to obtain initial scores and identify potential improvements. |
| 2 | [option2, option1, option1, option1] | (140, 145, 119) | Changes meal1 to option2 to substantially increase participant3's score while keeping participant1 and participant2 at reasonable levels, aiming to reduce score differences. |
| 3 | [option1, option2, option1, option1] | (133, 184, 156) | Modifies meal2 from option1 to option2 to further raise participant3's score, assuming this will reduce the gap between participants compared to previous plans. |
| 4 | [option1, option1, option2, option1] | (186, 163, 112) | Switches meal3 from option1 to option2 to boost participant3's preference on that meal, expecting a more balanced overall score distribution. |
| 5 | [option1, option1, option1, option2] | (210, 120, 44) | Changes meal4 from option1 to option2 to increase participant3's satisfaction on the last meal, aiming to compensate for consistently lower scores. |

Table 2: Search rounds of the non-reasoning model (Gemini-2.5-Flash without thinking), including plans, participant scores, and brief reasoning summaries.

## D.2 REASONING MODELS

Reasoning models generally perform better due to explicit search justifications, but they may still fail in more complex settings. We present a representative failure case from the 4-decision, 3-option scenario. Figure 9 shows the full search trajectory produced by the reasoning model, and Table 3 summarizes each round's reasoning, plan, and resulting divergence.

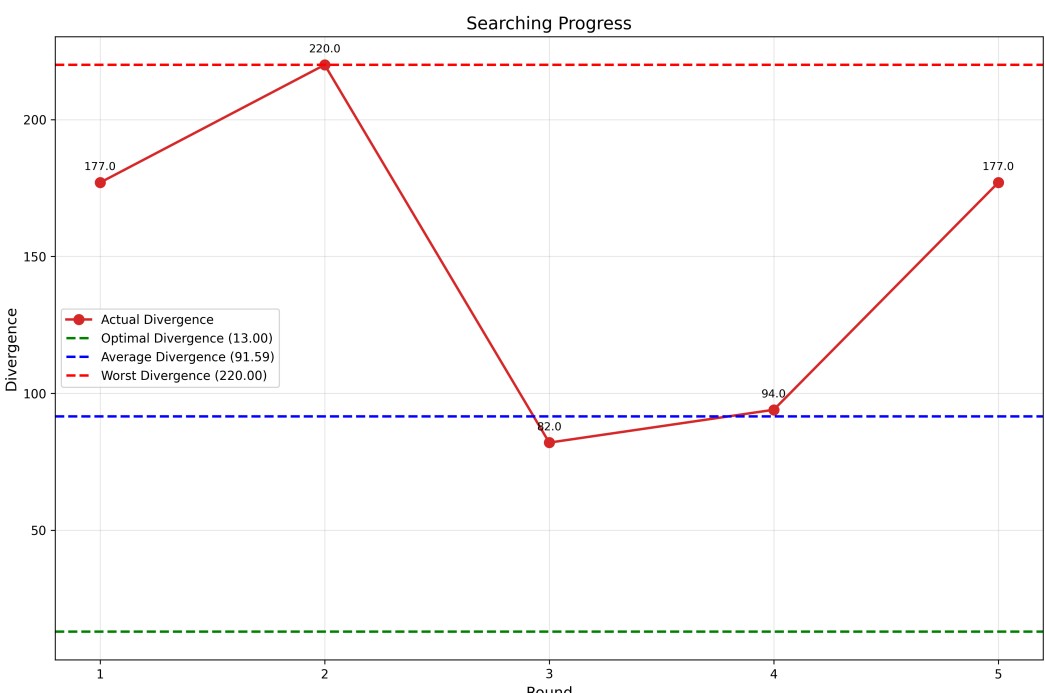

Figure 9: Search trajectory of the reasoning model in the 4-decision, 3-option setting.

| | Plan | Scores (p1, p2, p3) | Reasoning Summary |
|---|---|---|---|
| 1 | [oatmeal, salad, roast chicken, fruit] | (118, 249, 295) | Selected a balanced set across categories as an initial plan to minimize score differences. |
| 2 | [oatmeal, salad, stir-fry, fruit] | (69, 216, 289) | Modified dinner to boost player2's lower total and improve score balance. |
| 3 | [scrambled eggs, salad, roast chicken, fruit] | (138, 163, 220) | Adjusted breakfast and dinner to raise player2's satisfaction while narrowing cross-player gaps. |
| 4 | [pancakes, sandwich, steak, chips] | (150, 112, 56) | Introduced new options and changed all meals in an attempt to rebalance totals. |
| 5 | [oatmeal, salad, roast chicken, fruit] | (118, 249, 295) | Returned to the initial plan, incorrectly identifying it as the fairest option. |

Table 3: Search rounds of the reasoning model (GPT-o3-mini), including plans, participant scores, and brief reasoning summaries.

The failure of the reasoning model can be attributed to two primary factors. First, the model becomes trapped in a local optimum: although it discovers a plan with significantly lower divergence (Round 3), it fails to recognize it as the global best and later returns to inferior plans. Second, the model exhibits overly conservative adjustment behaviors. Instead of systematically exploring the solution space, it makes incremental or superficial modifications that do not meaningfully reduce divergence, and occasionally overcorrects without grounding its decisions in quantitative outcomes. Together, these issues lead the model to generate plausible-sounding reasoning while ultimately converging on suboptimal and less fair solutions.

# E  EXPERIMENTAL TUNING ANALYSES

This appendix provides additional details on the tuning procedures conducted during our experiments. We focus on two key components that influence search performance and computational efficiency, namely the number of search rounds used during iterative refinement and the prompt design that guides the LLM-based coordinator. These analyses inform the choices used in our final experimental setting.

## E.1  TUNING THE NUMBER OF SEARCH ROUNDS

To examine how the depth of the search procedure affects solution quality, we conducted experiments under the score feedback setting by varying the number of iterations to 3, 5, 7, and 9 rounds. The results reveal a clear pattern: increasing the number of iterations generally improves the likelihood of discovering higher quality plans.

When the iteration budget is limited to only three rounds, the coordinator often fails to sufficiently explore the search space, which frequently results in unsatisfactory solutions. Increasing the budget to five rounds yields substantial improvements, indicating that a moderate amount of exploration is necessary for the LLM to revise earlier decisions and escape poor initial choices.

Further expanding the search to seven or nine rounds produces only marginal additional gains. As the decision problem becomes more complex, these improvements diminish, while the associated token consumption and computational cost grow nearly proportionally with the number of iterations. Based on these observations, we adopt five rounds in our experimental setting because it provides a strong balance between achievable performance and computational efficiency.

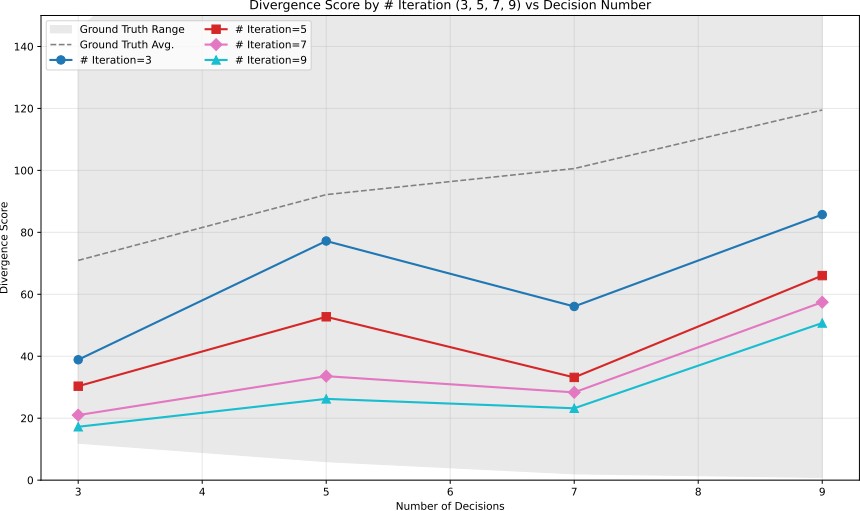

Figure 10: Comparison of performance across different numbers of search iterations. Increasing the iteration budget yields limited improvement beyond a certain point, while token usage and runtime continue to grow. We therefore select five iterations as the default parameter for all experiments.

## E.2  PROMPT DESIGN TUNING AND ABLATION

We also investigated the robustness of our prompt design and the contribution of individual components through an ablation study conducted under the score feedback setting. The full prompt contains three elements: a detailed contextual description that includes the explicit optimization objective, an illustrative example that demonstrates the expected reasoning pattern, and the search history from previous rounds.

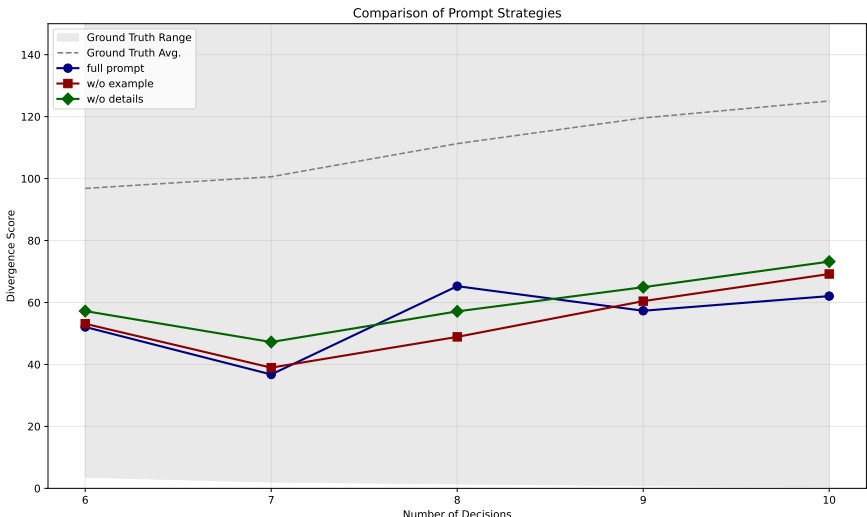

Figure 11: Effect of different prompt configurations on the LLM coordinator. We evaluate the impact of simplifying the context and removing the example. The full prompt performs best in most cases, and the context component contributes the most to the coordinator's effectiveness.

To assess sensitivity to different prompt formulations, we evaluated two degraded variants. The first variant removes the example, and the second variant simplifies the context by removing the explicit optimization objective, replacing it with a general instruction such as "find a fairer plan". Five independent test cases were conducted for each prompt configuration.

Across most experiments, the full prompt achieves the best performance, demonstrating its stability. Among the ablations, simplifying the context leads to the most pronounced degradation, which highlights the importance of explicitly specifying the optimization objective to guide the coordinator's reasoning process. Removing the example also results in weaker performance, although the impact is smaller. This suggests that examples are beneficial for shaping search behavior, but clear objective specification is the primary factor that determines solution quality.

Overall, these findings validate the prompt used in our experimental setting and highlight the importance of precise and structured guidance for effective LLM-driven search.

