# OpenReview forum: "Investigating Language Models for Supporting Complex Group Decisions"
_ICLR.cc/2026/Conference — Submitted to ICLR 2026_

### Official Review · Reviewer_rzJ5 · 2025-10-17

**Soundness:** 3
**Presentation:** 3
**Contribution:** 3
**Rating:** 6
**Confidence:** 5

**Summary:**

This paper investigates the use of Large Language Models (LLMs) to facilitate group decision-making processes where multiple participants need to reach consensus while balancing fairness and diverse preferences. The authors model group decisions as optimization problems over a decision surface with multiple options, where each participant receives gains from different choices, and the goal is to equalize participants' total profits.

The study explores various experimental conditions: (1) whether participant gains are private arbitrary numbers or public ordered sequences, (2) whether exact gains are known or hidden, and (3) whether communication is numeric or natural language-based. The LLM agent coordinates group decisions, optionally augmented with constraint solvers (CPLEX) or reinforcement learning (MCTS).

Key findings include: reasoning models (e.g o1-preview, o1-mini) significantly outperform non-reasoning models; integrating CPLEX or MCTS improves decision quality; numeric feedback yields better outcomes than verbal feedback; and the approach generalizes to rank-based preferences. The work demonstrates both the promise and limitations of LLMs in facilitating consensus for numerically-grounded group decisions.

**Strengths:**

**Originality:** This paper addresses a novel and important problem at the intersection of AI, group decision-making, and fairness. The formulation of group consensus as an optimization problem over a decision surface with profit equalization as a fairness criterion is creative. The systematic comparison of reasoning vs. non-reasoning LLMs, numeric vs. verbal feedback, and the integration of constraint solvers (CPLEX) and MCTS represents an innovative multi-faceted approach.

**Quality:** The experimental design is rigorous and systematic, exploring multiple dimensions: gain structures (arbitrary vs. ordered), visibility (public vs. hidden), and communication modalities (numeric vs. verbal). The use of both simulated and real-world scenarios (grant allocation) provides practical grounding. The authors conduct proper statistical analysis with multiple trials and compare against reasonable baselines including random and greedy strategies.

**Clarity:** The paper is generally well-written with clear motivation and problem formulation. The experimental setup is explained systematically, and results are presented with appropriate visualizations (Figures 2-5). The progression from simple to complex scenarios helps readers understand the methodology.

**Significance:** This work addresses an increasingly important problem as AI systems are deployed to facilitate human collaboration and decision-making. The findings about reasoning models' superiority, the importance of numeric feedback, and the benefits of hybrid AI approaches (LLM+CPLEX/MCTS) provide valuable insights for both researchers and practitioners working on AI-assisted group coordination.

**Weaknesses:**

**Scalability Concerns:** The experiments are limited to relatively small groups (3-5 participants) and decision surfaces (5-10 options). Real-world group decisions often involve larger groups and more complex decision spaces. The computational complexity of CPLEX and MCTS may become prohibitive as scale increases. The paper would benefit from analysis of how performance degrades with increasing problem size.

**Fairness Criterion Limitations:** The paper exclusively uses profit equalization as the fairness criterion. However, fairness in group decision-making is multifaceted and context-dependent. Other notions like proportional fairness, maximin fairness, or procedural fairness may be more appropriate in different contexts. The choice of fairness criterion is not well justified, and alternative fairness definitions are not explored.

**Missing Ablations:** While the paper compares different LLMs and augmentations (CPLEX, MCTS), some key ablations are missing. For example: What is the contribution of the specific prompt design? How sensitive is performance to hyperparameters (temperature, MCTS iterations, etc.)? What role does the iterative refinement play versus one-shot decision-making?

**Questions:**

1. **Prompt Engineering:** How sensitive are results to the specific prompts used? Have you tried alternative prompt formulations? Can you include ablation studies on prompt design?

2. **Failure Cases:** Can you characterize scenarios where the LLM-based approach fails or performs poorly? Are there specific problem structures or group dynamics that are particularly challenging?

3. **MCTS and CPLEX Integration:** How exactly are CPLEX and MCTS integrated with the LLM? Are they used to verify LLM outputs, generate candidate solutions, or something else? More algorithmic detail would be helpful.

---

> ### Author Response · Authors · 2025-12-03
> **Response to Reviewer rzJ5**
>
> Thank you for appreciating the originality, quality, and clarity of the manuscript. We answer your specific questions below.
>
> **Q4.1:**
> Prompt Engineering: How sensitive are results to the specific prompts used? Have you tried alternative prompt formulations? Can you include ablation studies on prompt design?
>
> **R4.1:**
> To verify the stability of our prompt design and understand the contribution of each component, we conducted an ablation study focused on the score feedback setting. Our full prompt includes (i) a detailed context description, (ii) an illustrative example, and (iii) search history from previous rounds. We evaluated two degraded variants: one removes the example, while the other simplifies the context by omitting an explicit optimization objective (i.e., instead of specifying the precise target objective, the prompt only instructs the LLM to “find a fairer plan”).
>
> We carried out five independent test cases under each configuration. As shown in Figure 11 of the revised manuscript, the full prompt consistently achieves the best results across most experiments, demonstrating its robustness. Among the ablations, simplifying the context causes the largest performance drop, confirming the importance of explicitly communicating the optimization objective to the coordinator. Removing the example also degrades performance, but to a lesser extent, indicating that while examples help guide the search behavior, clear objective specification is the primary factor driving solution quality.
>
> ---
>
> **Q4.2: Failure Cases:**
> Can you characterize scenarios where the LLM-based approach fails or performs poorly? Are there specific problem structures or group dynamics that are particularly challenging?
>
> **R4.2:**
> The non-reasoning model is able to generate alternative candidate solutions, but it lacks the ability to interpret score changes and therefore cannot determine an appropriate direction for iterative improvement. In contrast, the reasoning model can adjust its search direction based on intermediate outcomes, yet it is prone to getting trapped in local optima and failing to maintain the globally best solution. We provide detailed analyses and illustrative examples of these failure cases in Appendix D.
>
> ---
>
> **Q4.3: MCTS and CPLEX Integration:**
> How exactly are CPLEX and MCTS integrated with the LLM? Are they used to verify LLM outputs, generate candidate solutions, or something else? More algorithmic detail would be helpful.
>
> **R4.3:**
> We have added a detailed description and pseudocode of the coordinator’s search procedure in Appendix C of the revised manuscript. Conceptually, our method follows an iterative structure: in each round, the LLM coordinator proposes a candidate solution, collects feedback from all participants, transforms the feedback into an estimated gain signal, and optionally invokes a constraint solver (MCTS or CPLEX) to refine the search direction. The coordinator then updates the solution based on the solver’s suggestion and decides whether to continue or terminate the search. In other words, CPLEX and MCTS can serve as mechanisms for generating candidate solutions during the search process.

---

### Official Review · Reviewer_kAwL · 2025-10-25

**Soundness:** 1
**Presentation:** 2
**Contribution:** 1
**Rating:** 2
**Confidence:** 5

**Summary:**

This paper investigates how agents can facilitate complex group decisions. The toy problem has a facilitator who must guide a group of participants to reach a consensus on a series of choices. The goal is to select a combination of options that results in the fairest possible outcome. The experiments control for different forms of communication, reasoning vs non-reasoning models, and using external tools like MCTS. The results show that reasoning models are superior, and even better with access to tools.

**Strengths:**

- Studying coordination and consensus in multi-agent systems is important.
- The writing is mostly clear (except for the formulation)

**Weaknesses:**

The problem formulation in Section 3 is sloppy. For example:

- Minimize (high - low) is very confusing, and would have a straightforward solution where they’re equal. This minimizes the difference between the maximum and minimum total gains among all participants. The goal is to make the participants' final gains as equal as possible. Then you proceed to say: “The goal of the optimizer is to minimize the maximum divergence across the participants.” This minimizes regret at the individual level. I understand the point, but I don’t think your formalization makes it clear.
- You mention $x_{jk} \geq 0$ and $x_{jk} \in$ {0, 1}. The first term is redundant, since it’s already either 0 or 1. Moreover, the formalization of “Constraints for decision j” would be much simpler with words. You’re just saying these variables are binary and only one can be 1, or basically a one-hot encoding.

While there is value in good toy problems, the one you use is quite elementary. I don’t think we can generalize any of the findings to other realistic setups. For example, the facilitator could solve the problem and tell the participants. The participants themselves could solve the problem. In my opinion, you need a task where you have guarantees that not a single agent involved can unilaterally solve the problem.

Besides the simplicity of the task, you’re also exploring what happens in simple conditions. It’s hard to see a contribution from testing reasoning vs no reasoning, etc. We have good priors for these! If the facilitator is a model, then there should be large experiments with human participants. What characteristics of the facilitator make it better at this task? If you test the help of MCTS, that won’t generalize.

Ultimately, the results appear to be straightforward, indicating that what we’re learning is trivial. Reasoning models are better at this task, which depends on planning, but that’s exactly what these models are better at. Moreover, providing tools like MCTS also helps, which is expected.

**Questions:**

See weaknesses

---

> ### Author Response · Authors · 2025-12-03
> **Response to Reviewer kAwL**
>
> **Q3.1:**
> While there is value in good toy problems, the one you use is quite elementary. I don’t think we can generalize any of the findings to other realistic setups. For example, the facilitator could solve the problem and tell the participants. The participants themselves could solve the problem. In my opinion, you need a task where you have guarantees that not a single agent involved can unilaterally solve the problem.
>
> **R3.1:**
> As we discuss in more detail in general response G1, linear (integer) programming is an abstraction of many real-life decision problems. The generalization to integer programming allows us to explore problems where optimal solutions are difficult to find. We welcome further discussion on this topic.
>
> ---
>
> **Q3.2:**
> Besides the simplicity of the task, you’re also exploring what happens in simple conditions. It’s hard to see a contribution from testing reasoning vs no reasoning, etc. We have good priors for these! If the facilitator is a model, then there should be large experiments with human participants. What characteristics of the facilitator make it better at this task? If you test the help of MCTS, that won’t generalize. Ultimately, the results appear to be straightforward, indicating that what we’re learning is trivial. Reasoning models are better at this task, which depends on planning, but that’s exactly what these models are better at. Moreover, providing tools like MCTS also helps, which is expected.
>
> **R3.2:**
> Besides the generalization to the NP-hard integer programming problem, another important facet that we explore is when the participants do not reveal their preferences. As discussed in general response G3, the coordinator may not have all the information. During the interaction process, we expect LLMs to infer participants’ underlying preferences and reason and search for a solution that is as fair as possible for all participants. This is reminiscent of the hidden profile problem in psychology where groups systematically fail to share and integrate unique (unshared) information, leading them to make inferior decisions, even though the correct choice would be obvious if all information were pooled [1,2,3]. We explore four such settings: Verbal Feedback, Score Feedback, Ranking-based Scores, Plan Comparison. This is discussed in Section 1 of the revised manuscript.
>
> ---
>
> [1] Garold Stasser and William Titus. Pooling of unshared information in group decision making: Biased information sampling during discussion. J. Pers. Soc. Psychol., 48(6):1467–1478, June 1985.
>
> [2] Li Lu, Y Connie Yuan, and Poppy Lauretta McLeod. Twenty-five years of hidden profiles in group decision making: a meta-analysis. Pers. Soc. Psychol. Rev., 16(1):54–75, February 2012.
>
> [3] A variational-autoencoder approach to solve the hidden profile task in hybrid human-machine teams, N Pescetelli, P Reichert, A Rutherford, Plos one, 2022

---

### Official Review · Reviewer_X4di · 2025-10-30

**Soundness:** 2
**Presentation:** 2
**Contribution:** 2
**Rating:** 2
**Confidence:** 4

**Summary:**

The paper studies group decisions with a stylized mixed-integer program , a.k.a. menu planning. An LLM “planner” talks to three LLM “participants” for up to five rounds to pick options. The metric this paper used is “divergence” (max gap between participants’ totals). And the Main claims are the following: 1. Reasoning models beat non-reasoning; 2. Numeric feedback beats verbal and comparative; 3.  constraint solver or a reinforcement learning agent helps.

**Strengths:**

1. Section 5 experiments are clearly laid out;
2. plots make the patterns easy to see.
3. Limitations are acknowledged (LLM-only, need human studies and honesty assumptions).

**Weaknesses:**

1. Objective function is underspecified. Sec. 3 says “Minimize (high − low)” with constraints, but “high” and “low” are not defined semantically (are they max/min (V_i)? bounds? learned?). Add clear definitions and intuition.
2. Over-claiming without context. Statements like “reasoning models perform better” and “numeric feedback is better” are presented as broad truths, but when I got into it, the evidence comes from one stylized domain and one metric. Narrow the scope of the claims would help.
3. Stylized testbed only. The whole study uses one toy MIP “menu” template. No other domains, no real data. Claims should not generalize beyond this sandbox. Do the “reasoning > non-reasoning” and “numeric > verbal” results hold in another domain (not menu/MIP)?
4. Agents are all LLMs, and NO Humans!. Both planner and “participants” are LLMs; participants are fixed to GPT-4o-mini. Results may change with stronger or different models, and we do not learn about humans!
5. Planner models are a small set (e.g., o3-mini, Gemini-2.5-Flash, DeepSeek-R1/V3). The paper does not test stronger “thinking” baselines or ablate planning depth/rounds.

**Questions:**

1. Why is average divergence the only metric? Does it track social welfare or fairness well? Any correlation with Pareto distance? Can the authors try other metrics like welfare/utility, Pareto distance, envy, regret, stability, or success-rate metrics, other than the divergence?
2. Can you make more rigorous framing in places like Sec. 3 where you put “Minimize (high − low)”? What exactly are high and low in the objective? Please define and justify.
3. CPLEX gets full information and unsurprisingly wins. This is not a fair operational baseline for real settings where scores are hidden. Maybe frame it as an upper bound, and report optimality gap?
4. How many search rounds matter? You cap at five. But what is the trade-off with more rounds?

---

> ### Author Response · Authors · 2025-12-03
> **Response to Reviewer X4di**
>
> **Q2.1:**
> Why is average divergence the only metric? Does it track social welfare or fairness well? Any correlation with Pareto distance? Can the authors try other metrics like welfare/utility, Pareto distance, envy, regret, stability, or success-rate metrics, other than the divergence?
> Can you make more rigorous framing in places like Sec. 3 where you put “Minimize (high − low)”? What exactly are high and low in the objective? Please define and justify.
>
> **R2.1:**
> We have clarified the use of divergence in general response G3. We have also added a clarification in Section 3 of the manuscript. High is the maximum possible value of divergence (amongst all plans) and Low is the smallest possible value. The region in grey shows the feasible space.
>
> Thank you for pointing out alternative objective functions. We ran experiments with the measure of welfare [5]. The results are summarized in general response G3, section 5.6, and Figure 7 of the revised manuscript.
>
> ---
>
> **Q2.2:**
> CPLEX gets full information and unsurprisingly wins. This is not a fair operational baseline for real settings where scores are hidden. Maybe frame it as an upper bound, and report optimality gap?
>
> **R2.2:**
> Thanks for the suggestion. We now report CPLEX values as an upper bound and show this in Figure 2 of the revised manuscript.
>
> ---
>
> **Q2.3:**
> How many search rounds matter? You cap at five. But what is the trade-off with more rounds?
>
> **R2.3:**
> Thanks for the suggestion. We have conducted new experiments to determine the effect of the number of search rounds. We further evaluated the performance under the same score feedback setting by varying the number of iterations (3, 5, 7, and 9 search problems). The results show that increasing the number of iterations generally improves the likelihood of discovering better plans. When the iteration budget is limited to only 3 rounds, the coordinator often fails to reach a satisfactory solution due to insufficient exploration. However, although performance improves when increasing to 5 iterations, further extending the search to 7 or 9 iterations yields only marginal additional gains, particularly as the problem complexity grows. At the same time, the longer search introduces significantly higher token usage and computational overhead. Therefore, we selected 5 iterations as the optimal trade-off, achieving most of the attainable performance improvement while maintaining a reasonable cost. This is discussed in Appendix E.1 of the revised manuscript.

---

### Official Review · Reviewer_ZWeN · 2025-10-30

**Soundness:** 2
**Presentation:** 2
**Contribution:** 2
**Rating:** 2
**Confidence:** 3

**Summary:**

This paper studies how the current LLMs approach the problem of complex group decisions, which is formulated as a mixed-integer programming problem.

**Strengths:**

The problem setting is novel.

**Weaknesses:**

The technical contribution is limited.

**Questions:**

I am generally concerned about the motivation of this paper: it seems to evaluate the capability of LLMs on solving one particularly type of mixed-integer programming problems. However, why is this type of particular interest compared with others in terms of evaluating LLMs' capability? Also, the methodologies being evaluated with are also seems to be limited: direct prompting or MCTS. I would imagine a more reasonable way for LLMs to solve this kind of tasks is to provide a set of tool calls which can directly solve the tasks via optimization procedure.

---

> ### Author Response · Authors · 2025-12-03
> **Response to Reviewer ZWeN**
>
> Thank you for your evaluation and feedback on the paper.
>
> **Q1.1:** I am generally concerned about the motivation of this paper: it seems to evaluate the capability of LLMs on solving one particular type of mixed-integer programming problems. However, why is this type of particular interest compared with others in terms of evaluating LLMs’ capability?
>
> **R1.1:** Our specific decision problem is motivated by three considerations. First, linear (or integer) programming is essential in decision making as it provides a rigorous, quantitative method for choosing the best possible course of action under constraints. By expressing objectives and encoding real-world limitations as inequalities, many potential solutions can be evaluated to identify the optimal one. This generic characterization is especially useful in complex environments such as supply chains, scheduling, finance, energy systems, and public policy.  Second, integer programming is a natural generalization of linear programming that is NP-hard. This means that any algorithm (or human being) can at best produce an approximation. So, even in environments where tools may assist LLMs, approximations to the ideal solution can be compared for how well they satisfy the objective and how efficient they are. Finally, participants in decision making may hold private preferences that are not shared with others, and the only hint that the group may have is their degree of satisfaction in a natural language response. During the interaction process, we expect LLMs to infer participants’ underlying preferences and reason and search for a solution that is as fair as possible for all participants. This is reminiscent of the hidden profile problem in psychology where groups systematically fail to share and integrate unique (unshared) information, leading them to make inferior decisions, even though the correct choice would be obvious if all information were pooled [1,2,3]. This is discussed in Section 1 of the revised manuscript.
>
> ---
>
> **Q1.2:** Also, the methodologies being evaluated with are also seems to be limited: direct prompting or MCTS. I would imagine a more reasonable way for LLMs to solve this kind of tasks is to provide a set of tool calls which can directly solve the tasks via optimization procedure.
>
> **R1.2:** The incorporation of CPLEX and MCTS is our investigation of how and when an LLM should use auxiliary tools.  We have explained the different design choices in general responses G2, G3, and G4. The benefit from auxiliary tools depends on the specific kind of information that the LLM/coordinator is provided. We have considered five different possibilities: Complete Information, Verbal Feedback, Score Feedback. Ranking-based Scores, and Plan Comparison. In the revised draft, we have also considered other prompting strategies. Please see Response R4.1 to Reviewer rzJ5.
>
> ---
>
> [1] Garold Stasser and William Titus. Pooling of unshared information in group decision making: Biased information sampling during discussion. J. Pers. Soc. Psychol., 48(6):1467–1478, June 1985.
>
> [2] Li Lu, Y Connie Yuan, and Poppy Lauretta McLeod. Twenty-five years of hidden profiles in group decision making: a meta-analysis. Pers. Soc. Psychol. Rev., 16(1):54–75, February 2012.
>
> [3] A variational-autoencoder approach to solve the hidden profile task in hybrid human-machine teams, N Pescetelli, P Reichert, A Rutherford, Plos one, 2022

---

### Author Response · Authors · 2025-12-03
**General response to all reviewers**

We thank the reviewers for their feedback on the paper. Here, we would like to address some general questions that have arisen in more than one review.

**G1. Problem motivation:**

Our specific decision problem is motivated by three considerations. First, linear (or integer) programming is essential in decision making as it provides a rigorous, quantitative method for choosing the best possible course of action under constraints. By expressing objectives and encoding real-world limitations as inequalities, many potential solutions can be evaluated to identify the optimal one. This generic characterization is especially useful in complex environments such as supply chains, scheduling, finance, energy systems, and public policy.  Second, integer programming is a natural generalization of linear programming that is NP-hard. This means that any algorithm (or human being) can at best produce an approximation. So, even in environments where tools may assist LLMs, approximations to the ideal solution can be compared for how well they satisfy the objective and how efficient they are. Finally, participants in decision making may hold private preferences that are not shared with others, and the only hint that the group may have is their degree of satisfaction in a natural language response. During the interaction process, we expect LLMs to infer participants’ underlying preferences and reason and search for a solution that is as fair as possible for all participants. This is reminiscent of the hidden profile problem in psychology where groups systematically fail to share and integrate unique (unshared) information, leading them to make inferior decisions, even though the correct choice would be obvious if all information were pooled [1,2,3]. This is discussed in Section 1 of the revised manuscript.

**G2. Privacy constraints:**

We considered three different settings for personal preferences by agents:

1. Complete Information: Participants reveal their preferences explicitly. This setting evaluates the ability of a language model to solve a mixed integer programming problem. The problem is NP-hard and we do not expect the language model to find the optimal answer but it is a test of its capability of searching in high dimensional spaces under constraints

2. Verbal Feedback: Each participant verbalizes their preference in natural language. The use of an LLM in translating such preferences into numbers is a crucial test of their ability.

3. Score Feedback: Participants only report the overall score of the current plan. This measures an LLM’s ability to find plans (combinations of decisions) to evaluate in order to converge towards an optimal solution.

4. Ranking-based Scores. Each participant only maintains only a relative ranking of options within a decision instead of specific scores. Their feedback is based only on this relative order. This is a simplification of preferences (numeric values) to ordinals.

5. Plan Comparison. Instead of asking agents to evaluate a single candidate plan, the planner asks them to rank order two plans in order to infer a gradient in the plan space.

These protocols are detailed in Section 4.3 of the manuscript.

---

> ### Author Response · Authors · 2025-12-03
>
> **G3. Objective function:**
>
> The manuscript considered the objective function of ``divergence’’ that is a measure of fairness. Every participant has a gain based on a proposed plan. The goal of the planner is to ensure that the gap between the maximum and the minimum gains is minimized. This measure is different from other notions of optimality such as envy-fairness that are based on distribution of resources [4]. We can consider the gain of each participant as an objective and analyze the corresponding multi-objective criterion for optimality. However, the optimal plan in terms of divergence is not pareto-optimal: the gain of a specific participant may be higher for a non-optimal plan.
>
> Based on the reviewers’ feedback (specifically Reviewer Zndi),  we implemented another objective function, group welfare, motivated by [5]. In our setting, welfare is defined as the sum of all participants’ individual gains under a given plan. Accordingly, we updated the LLM coordinator’s prompt to explicitly instruct the planner to search for solutions that maximize the total group benefit. Under this setup, we compared the performance of score feedback and verbal feedback strategies. The results show that when the decision dimensionality is low, both strategies perform similarly. However, as the number of decisions increases and the search space expands, verbal feedback consistently outperforms score feedback, achieving substantially higher welfare.
>
> Through detailed case analysis, we find that this performance gap stems from the fundamentally different search behaviors induced by the two feedback modalities. When receiving score feedback, the LLM coordinator tends to adopt a greedy and conservative local search strategy, typically modifying only one or two decisions at a time and reverting changes that lower the welfare. This risk-averse behavior makes it more prone to getting trapped in local optima in high-dimensional search spaces. In contrast, verbal feedback provides richer contextual information (e.g., which options are “positively received” or “consistently negative”), enabling the coordinator to perform more assertive and globally informed exploration by simultaneously altering multiple decisions or introducing previously untested options. As a result, verbal feedback is more capable of escaping local optima and identifying higher-welfare solutions when the decision space grows. This discussion appears in Section 5.6 of the revised manuscript and the corresponding plot is in Figure 7.
>
>
> **G4. Use of auxiliary tools:**
>
> We explored the use of specific tools that can aid the planner in finding the optimal decision. We examined two such tools: a constraint solver (CPLEX) and an RL strategy (MCTS). Together these designs consider the question of how specialized tools can assist an LLM in solving and interacting with decision-making groups.
>
> [1] Garold Stasser and William Titus. Pooling of unshared information in group decision making: Biased information sampling during discussion. J. Pers. Soc. Psychol., 48(6):1467–1478, June 1985.
>
> [2] Li Lu, Y Connie Yuan, and Poppy Lauretta McLeod. Twenty-five years of hidden profiles in group decision making: a meta-analysis. Pers. Soc. Psychol. Rev., 16(1):54–75, February 2012.
>
> [3] A variational-autoencoder approach to solve the hidden profile task in hybrid human-machine teams, N Pescetelli, P Reichert, A Rutherford, Plos one, 2022
>
> [4] Brams, S. J., & Taylor, A. D. (1996). Fair Division: From Cake-Cutting to Dispute Resolution.
> Cambridge University Press.
>
> [5] Harsanyi, John C. "Cardinal welfare, individualistic ethics, and interpersonal comparisons of utility." Journal of political economy 63.4 (1955): 309-321.

---

### Meta-Review · Area_Chair_1X79 · 2026-01-05

**Summary:**

Some reviewers believed that the paper tries to tackle an important problem. The main concern is that most reviewers are unconvinced by the proposed approach, due to simplicity of the tasks,  straightforward conclusions, over-claiming the results, and no human in experiments. While one reviewer was more positive, the overall sentiment was quite negative

**Reviewer Concerns:**

see above

**Reviewer Scores:**

NA

---

### Decision · Program_Chairs · 2026-01-26

Reject